# Formyl Peptide Receptor 2 Alleviates Hepatic Fibrosis in Liver Cirrhosis by Vascular Remodeling

**DOI:** 10.3390/ijms22042107

**Published:** 2021-02-20

**Authors:** Ji Hye Jun, Soo Young Park, Sohae Park, Hee Jung Park, Jae Yeon Kim, Gyu Tae Park, Si Hyun Bae, Jae Ho Kim, Gi Jin Kim

**Affiliations:** 1Department of Biomedical Science, CHA University, Seongnam 13488, Korea; jihyejun1015@gmail.com (J.H.J.); stock75@snu.ac.kr (S.Y.P.); sohae11@snu.ac.kr (S.P.); heejung970328@gmail.com (H.J.P.); janejaeyeon92@gmail.com (J.Y.K.); 2Department of Physiology, School of Medicine, Pusan National University, Yangsan 50612, Korea; daramzuy2@naver.com; 3Department of Internal Medicine, Catholic University Medical College, Seoul 03312, Korea; baesh@catholic.ac.kr

**Keywords:** liver cirrhosis, WKYMVm, angiogenesis, fibrosis, regeneration

## Abstract

Hexapeptide WKYMVm (Trp-Lys-Tyr-Met-Val-D-Met), a ligand of formyl peptide receptor 2, exhibits anti-inflammatory and angiogenic properties in disease models. However, the therapeutic effects of WKYMVm on hepatic fibrosis have not been evaluated to date. Therefore, we investigated whether WKYMVm exerts antifibrotic effects and induces vascular regeneration in a rat model of bile duct ligation (BDL). The antifibrotic and angiogenic effects of WKYMVm on liver regeneration in the BDL rat model were analyzed using biochemical assays, qRT-PCR, western blotting, immunofluorescence, and immunohistochemistry. To determine the effects of WKYMVm on hepatic fibrosis and angiogenesis in vitro, we measured the expression levels of fibrotic factors in hepatic stellate cells (HSCs) and angiogenic factors in human umbilical vein endothelial cells (HUVECs). WKYMVm attenuated the expression of collagen type I (Col I) and α-smooth muscle actin (α-SMA) and significantly increased the levels of angiogenetic factors in the BDL model (*p* < 0.05). WKYMVm reduced fibrotic marker expression in transforming growth factor (TGF)-β-induced HSCs and promoted angiogenic activity through tube formation in 5-Fluorouracil (FU)-treated HUVECs (*p* < 0.05). Also, WKYMVm administration enhanced hepatocyte proliferation in BDL rats (*p* < 0.05). The WKYMVm alleviates hepatic fibrosis by inhibiting HSC activation and promotes hepatic regeneration via vascular remodeling. These data suggest that the WKYMVm may be a new therapeutic agent for liver fibrosis.

## 1. Introduction

Hepatic failure induced by chronic liver injury is related to liver diseases that are difficult to cure and is associated with high morbidity and mortality [1]. Chronic liver disease involves inflammation, activation of HSCs, which causes fibrogenesis, and necrosis of hepatocytes caused by vascular blockade [2]. These processes lead to hepatic microvascular changes, deposition of extracellular matrix (ECM), and hepatic endothelial dysfunction [3]. Liver fibrosis is characterized by excessive deposition of ECM components along with acute and chronic inflammation [4]. In the normal liver, HSCs exist in a quiescent non-proliferative state; however, HSCs exposed to acute and chronic liver injury are activated by profibrogenic cytokines and transdifferentiated into myofibroblasts that express collagen and α-SMA, thereby contributing to hepatic fibrogenesis [5,6,7].

Specifically, hepatic fibrogenesis impairs the vasculature of the liver and simultaneously induces liver sinusoidal endothelial cell (LSEC) dysfunction [8]. The hepatic vascular niche, which is mainly represented by LSECs, secretes angiogenic factors to stimulate hepatic regeneration [9]. Among these angiogenic factors, VEGF is a potent regulator of angiogenesis and vascular remodeling due to its effects on endothelial cell (EC) survival and proliferation [10]. It has been shown that the VEGF expression is markedly increased during liver regeneration induced either by partial hepatectomy or drug intoxication [11]. The biological effects of VEGF can be explained by two tyrosine kinase receptors, namely, VEGF receptor 1 (VEGFR1) and VEGF receptor 2 (VEGFR2), which participate in different signaling pathways [12]. Activation of VEGF receptors (VEGFRs) results in paracrine release of hepatocyte growth factor (HGF), interleukin-6 (IL-6), and other hepatotropic molecules from LSECs and leads to EC regeneration [13]. In a mouse model with liver fibrosis, expression of HGF in ECs prevents the activation of perivascular fibroblasts and serves as a key molecule for preventing fibrosis [14]. This process is considered advantageous for tissue regeneration as well as restoration of injured tissues [12]. In particular, the Wnt/β-catenin pathway plays an important role in angiogenesis and vessel remodeling. Wnt/β-catenin signaling can promote EC proliferation and induce cell cycle progression through transcriptional activation of cyclin D1 in ECs [15]. Additionally, frizzled 4 and 7, which are upstream of β-catenin, are regulated by VEGF and control endothelial permeability and integrity by interacting with VE-cadherin and β-catenin [16,17,18].

The synthetic peptide WKYMVm (Trp-Lys-Tyr-Met-Val-D-Met) is a strong agonist of formyl peptide receptor 2 (FPR2), which is a transmembrane protein belonging to the G protein-coupled receptor family [19]. WKYMVm has been reported to exert therapeutic effects in various disease models; for example, WKYMVm promotes neovascularization and tissue repair through the migration and proliferation of endothelial colony-forming cells (ECFCs) or circulating angiogenic cells (CACs) in models of hind limb and myocardial ischemia [20,21]. Additionally, topical application of WKYMVm promotes cutaneous wound healing in a streptozotocin-injured diabetic model through the formation of von Willebrand factor (vWF)-positive vessels and re-epithelialization [22]. The WKYMVm has been known to also bind the hepatocyte growth factor receptor (HGFR, tyrosine-protein kinase Met, c-Met) (www.nextprot.org). However, the therapeutic effects of WKYMVm in hepatic failure have not yet been studied.

Therefore, our objectives are to examine the effects of WKYMVm on hepatic regeneration via vascular remodeling, resulting from its proangiogenic properties in a rat model of BDL-induced hepatic fibrosis.

## 2. Results

### 2.1. WKYMVm Attenuates Liver Fibrosis in the BDL Rat Model

The structure of WKYMVm was shown in Figure 1A. The peptide WKYMVm is composed to Trp-Lys-Tyr-Met-Val-D-Met as hexapeptide. As the first residue of WKYMVm is tryptophane, the half-life of WKYMVm isknown to be proportionally short. As shown in Figure 1B, the WKYMVm binds with formyl peptide receptor 2 (FPR2), which is the main receptor of WKYMVm. To evaluate whether FPR2 and hepatocyte growth factor receptor (HGFR, c-Met, tyrosine-protein kinase Met), which is the receptor of WKYMVm, is expressed in the rat liver, mRNA level of FPR2 and HGFR was analyzed by qRT-PCR. The FPR2 and HGFR were expressed in basal level in normal rat liver; however, the expression was decreased in BDL non-treated (NTx) group, while their level was significantly increased in WKYMVm treated BDL (+WK) group (* *p* < 0.05; Figure 1C,D). To evaluate the therapeutic effect of WKYMVm peptide treatment in a liver injury rat model, serological analysis was performed. Hepatic function was obviously impaired in the NTx BDL group based on the serum levels of alanine aminotransferase (ALT), aspartate aminotransferase (AST), total bilirubin, and albumin (ALB), whereas treatment with WKYMVm restored the serum levels of these indicators similar with those observed in the normal group (* *p* < 0.05; Figure 1E–H). Histological changes related to liver injury and fibrosis wereanalyzed in liver tissues stained with Hematoxylin and Eosin (H&E) and Sirius Red. Compared with normal liver tissues, the liver tissues of the BDL group exhibited distorted architecture and an increase in the portal vein diameter that depended on the degree of fibrosis (data not shown). As determined by Sirius Red staining, the NTx group exhibited remarkable collagen accumulation in liver tissues. However, WKYMVm treatment markedly attenuated fibrotic deposition and then decreased the portal vein diameter and collagen accumulation in fibrotic tissues (* *p* < 0.05; Figure 1J).

Generally, the expression of type I collagen (Col I) and alpha-smooth muscle actin (α-SMA) are used as representative factors for evaluating hepatic fibrosis. So, we demonstrated that their expression level by qRT-PCR and western blotting, respectively. As shown in Figure 2A,B, the mRNA expression was shown increased tendency in NTx group versus normal control. Also, the WKYMVm treatment had eased off mRNA levels of fibrotic markers. Protein expression level of Col I in the +WK group was significantly decreased compared to those of the +WK group (* *p* < 0.05; Figure 2C). In addition, α-SMA expression were significantly downregulated in the +WK group compared with the NTx group (* *p* < 0.05; Figure 2B,D). These data indicate that WKYMVm attenuates hepatic dysfunction and fibrosis induced by BDL in the rat liver.

### 2.2. WKYMVm Inhibits Hepatic Stellate Cell Activation Induced by Transforming Growth Factor-Beta (TGF-β)

During the progression of liver fibrosis, hepatic stellate cell (HSC) is a major cell type responsible for fibrosis and are activated by the stimulation of profibrogenic cytokines, including transforming growth factor-beta (TGF-β). So, we hypothesized whether WKYMVm can suppress HSC activation in vitro. HSCs stimulated by TGF-β were cultured with or without WKYMVm. The expression of Col I and α-SMA were markedly upregulated in TGF-β-treated HSCs compared with normal HSCs, whereas treatment with WKYMVm significantly suppressed the upregulation of α-SMA expression although the effect of WKYMVm was not shown in Col I expression (* *p* < 0.05; Figure 2E,F). Notably, immunofluorescence analysis showed that WKYMVm reduced the expression of α-SMA in TGF-β-treated HSCs (Figure 2G). In the fluorescent quantitative data via immunofluorescence, the intensity of α-SMA was dramatically increased in TGF-β-treated HSCs compared with normal HSCs. In contrast to TGF-β-treated HSCs, WKYMVm treatment mitigated the accumulation of α-SMA (* *p* < 0.05; Figure 2H). These results suggest that WKYMVm has an inhibitory effect on fibrogenesis through the regulation of Col I and α-SMA expression in TGF-β-stimulated HSCs.

### 2.3. WKYMVm Promotes Vascular Remodeling in the BDL Rat Liver

During the progression of fibrosis in liver tissues, vascular structure abnormalities are common phenomenon. In previous reports, Kim and colleagues reported that WKYMVm has potential proangiogenic effects in the ischemic heart [21,23]. We confirmed that the irregular and increased diameter of portal vein by BDL was restored to control level by WKYMVm treatment (* *p* < 0.05, Figure 1I). To further investigate the effect of WKYMVm on vascular remodeling in the BDL rat liver, we measured the expression of angiogenic factors including vascular endothelial growth factor (VEGF), vascular endothelial growth factor receptor 1 (VEGFR1; Flt1), vascular endothelial growth factor receptor 2 (VEGFR2; Flk1/KDR), and endoglin (ENG; CD105) using qRT-PCR and western blot. Interestingly, the expression of angiogenic factors were significantly increased in the +WK group more than those of in the NTx group. Especially, VEGF as the key regulator of angiogenesis and vascular remodeling tightly regulates EC survival and proliferation. The mRNA and protein expression of VEGF was dramatically upregulated by WKYMVm treatment although the difference between Nor and NTx groups was not statistically significant (* *p* < 0.05, Figure 3A–C). Additionally, the effects of VEGF are mainly mediated through two receptors, namely, VEGFR1 and 2 [24]. The mRNA level of VEGFR1 showed increased tendency in NTx compared to Nor group, and its expression was significantly induced by WKYMVm treatment versus NTx (*p* < 0.05, Figure 3D). Also, VEGFR2, which has more binding affinity with VEGF, was repressed by BDL compared that normal liver. However, by WKYMVm treatment in BDL liver its level was evidently restored (* *p* < 0.05, Figure 3E). Endoglin is a marker of activated ECs and plays an important role in angiogenesis and vascular homeostasis. Their expression by WKYMVm treatment were significantly induced while the difference between Nor and NTx groups was not represented (* *p* < 0.05, Figure 3F). Furthermore, the expression and localization of von Willebrand factor (vWF), which is a marker for blood vessel formation, and α-SMA were analyzed using immunofluorescence. In the +WK group, vWF was strongly expressed in the intimal lining of portal blood vessel, as in the normal rat liver, while the NTx group showed a marked decrease in vWF expression and an increase in α-SMA expression. These data demonstrate that WKYMVm promotes vascular remodeling through activation of angiogenic factors in the BDL rat model.

### 2.4. WKYMVm Enhances Angiogenic Activity in HUVECs

Based on the observation of the vascular remodeling effects of WKYMVm in the BDL rat liver, we further determined the effect of WKYMVm on angiogenesis in vitro using HUVECs. HUVECs were pretreated with fluorouracil (5-FU), an anti-cancer drug and EC inhibitor, and then treated with WKYMVm. The mRNA expression levels of angiogenic factors such as VEGF, VEGFR1, and VEGFR2 were analyzed using qRT-PCR. WKYMVm treatment markedly increased the mRNA level of VEGF in 5-FU-treated HUVECs (* *p* < 0.05, Figure 4A). The expression of VEGFR1 and 2 was showed significantly decreased results in 5-FU-treated HUVECs compared to normal HUVECs, showing HUVECs are damaged by 5-FU. Also, by treatment of WKYMVm in HUVEC, their expressions were slightly restored compared to non-treated WKYMVm HUVECs although they didnot show statistical significance (* *p* < 0.05, Figure 4B,C). Also, secreted VEGF into culture supernatant was increased in 5-FU-treated HUVECs after WKYMVm treatment (* *p* < 0.05, Figure 4D). Moreover, the tube formation ability of HUVECs was significantly enhanced by WKYMVm treatment (* *p* < 0.05, Figure 4E,F). To examine the effect of WKYMVm on HUVEC permeability for evaluation of endothelial function, a dextran endothelial permeability assay was performed, as shown in the scheme in Figure 4G. The inhibitory effect of 5-FU in HUVECs led to a significant increase in endothelial permeability compared to that in normal HUVECs, whereas WKYMVm treatment decreased endothelial permeability in 5-FU-treated HUVECs although the statistical significance was not shown (Figure 4H). Also, the direct effect of FPR2 by WKYMVm was confirmed by siRNA-FPR2 transfection in HUVEC. In 5-FU treated and siRNA-FPR2 transfected HUVEC, tube formation and dextran permeability were significantly decreased and increased, respectively compared to only 5-FU treatment with HUVEC (* *p* < 0.05, Appendix AA–C). Previous studies have shown that the Wnt/β-catenin signaling pathway contributes to the regulation of vascular regeneration in the liver [25,26,27]. To analyze whether WKYMVm can affect Wnt/β-catenin signaling associated with angiogenesis, we stained active and non-phosphorylated β-catenin by immunofluorescence and evaluated the nuclear localization of β-catenin. As shown in Figure 4I, active and non-phosphorylated β-catenin expression was markedly lower in the cell nuclei of 5-FU-treated HUVECs than in those of normal HUVECs, while active β-catenin levels in 5-FU-treated HUVECs were significantly increased by WKYMVm treatment (* *p* < 0.05, Figure 4I,J). These results indicate that the WKYMVm peptide promotes angiogenic events and functions by upregulating the expression of VEGF and its receptors via Wnt/β-catenin signaling in HUVECs.

### 2.5. WKYMVm Improves Hepatic Regeneration in the BDL Rat Model

The liver regeneration mechanism is initiated by several growth factors and the IL-6/gp130/STAT3 signaling pathway [28]. To determine whether the administration of WKYMVm peptide induces hepatic regeneration in the BDL rat model, we analyzed the expression level of factors related to liver regeneration. Hepatocyte nuclear factor 1 (HNF1) is a transcription factor involved in the regulation of a large set of hepatic genes, including ALB [29]. The mRNA and protein levels of HNF1α and ALB weresignificantly increased in the +WK group compared with the NTx group (* *p* < 0.05, Figure 5A,B,D–F). Also, we analyzed the protein levels of interleukin-6 (IL-6), glycoprotein 130 (gp130), which is the type I cytokine receptor of IL-6, and phosphorylated STAT3 in the BDL rat liver. Administration of WKYMVm significantly promoted the expression of IL-6 and phosphorylated STAT3 in BDL rats (* *p* < 0.05, Figure 5D,G,H). However, in the +WK group, the protein expression of gp130 showed a slightdifference compared to theNTx group (Figure 5D and Appendix AB). The serum level of HGF was determined using ELISA. Serological HGF expression represented increased tendency in the +WK group compared to the NTx group although there was no statistical significance (Figure 5C).

To further examine the effect of the WKYMVm peptide on hepatocyte proliferation, we assessed PCNA expression in liver tissues by immunohistochemistry and quantified PCNA positive cells by using digital image analysis technology and cyclin D1 expression using western blot. As shown in Figure 5I,J, the number of PCNA-positive hepatocytes was significantly increased in the +WK group than in the NTx group (* *p* < 0.05, Figure 5I,J). Also, the protein expression of cyclin D1 was also increased in the +WK group than in the NTx group (Figure 5D and Appendix AA). These findings suggest that WKYMVm administration enhances hepatocyte proliferation and hepatic regeneration in BDL rats through IL-6/gp130/STAT3 signaling.

## 3. Discussion

Hepatic fibrosis is a common pathological process of chronic liver injuries of various etiologies and is characterized by a multicellular response along with the activation of HSCs [30]. In the present study, we first demonstrated that WKYMVm exerted therapeutic effects, including antifibrotic and hepatic regeneration effects, through vascular remodeling in a rat model of BDL-induced liver cirrhosis. WKYMVm improved the levels of serum ALB and hepatic enzymes, which are important indices of liver function. Additionally, WKYMVm restored hepatic architectural and deposition of collagen in the fibrotic liver tissue. HSCs activated by profibrogenic cytokines are considered the main fibrogenic cell type in hepatic injury and induce the expression of Col I and α-SMA. HSCs were markedly activated in the NTx group, as shown by upregulation of Col I and α-SMA expression by qRT-PCR, while WKYMVm treatment downregulated the expression levels of these factors (Figure 2A,B). It is well known that TGF-β is the most potent profibrogenic cytokine that contributes to HSC activation and the accumulation of ECM in the progression of liver fibrosis. In vitro, WKYMVm inhibited the expression of α-SMA in the rat HSCs stimulated with TGF-β (* *p* < 0.05; Figure 2F–H). Our in vivo and in vitro studies showed that WKYMVm has an inhibitory effect against hepatic fibrosis (* *p* < 0.05; Figure 2).

Angiogenesis plays an important role in many physiological and pathological processes associated with hepatic regeneration. HSCs and ECs can promote pathological angiogenesis by expressing a variety of angiogenic factors [31]. Although VEGF is a key regulator of angiogenesis and vascular remodeling and a survival factor for EC, VEGF is upregulated and improves sinusoidal reconstruction during liver regeneration in partial hepatectomy [32]. Expression of VEGF and VEGFRs are correlates with the rate of sinusoidal EC (SEC) proliferation after acute hepatic failure [9]. Recently, Yang et al. reported that VEGF has a dual and opposing role in fibrogenesis and fibrosis resolution through the critical effects of VEGF on vascular permeability [10]. WKYMVm was previously shown to exert proangiogenic effects in the ischemic heart [21]. In the present study, we found that WKYMVm induces upregulation of VEGF and VEGFR expression in vivo and in vitro (* *p* < 0.05; Figure 3A–E and Figure 4A–D). Furthermore, WKYMVm treatment enhanced angiogenic activities, as evidenced by increased tube formation and decreased EC permeability in vitro (* *p* < 0.05; Figure 4E–H). It has been reported that the Wnt/β-catenin signaling pathway can affect downstream gene expression of angiogenic factors, including VEGF and matrix metalloproteinases (MMPs) [25]. The present results suggested that WKYMVm treatment leads to increased localization of β-catenin in the nucleus, where it serves as a transcription factor to activate downstream target genes (* *p* < 0.05; Figure 4I,J). Therefore, WKYMVm may effectively enhance angiogenic activity via regulation of the Wnt/β-catenin signaling pathway.

Additionally, Hu J and his colleagues were reported that the WKYMVm peptide is known to inhibit osteoclastogenesis by downregulating the expression of inflammatory cytokines such as IL-1β, tumor necrosis factor-alpha (TNF-α), andinducing the phosphorylation of signal transducer and activator of transcription 3 (STAT3) via CD9/gp130/STAT3 signaling [33]. In our study, we also confirmed that phosphorylation of STAT3 is induced by WKYMVm treatment in the livers of BDL model (* *p* < 0.05; Figure 5D,H). The phosphorylation of STAT3 and the expression of IL-6 and gp130 were increased in the livers of BDL rats by WKYMVm treatment. STAT3 also has been reported in the other studies that it is involved in regeneration through the IL-6/gp130/JAK pathway in hepatic-specific processes [34,35]. Additionally, we showed that the peptide WKYMVm promotes the proliferation of hepatocytes in the liver via the IL-6/gp130/STAT3 signaling pathway, a signaling pathway that is associated with liver regeneration (* *p* < 0.05; Figure 5 and Appendix A).

However, the peptide WKYMVm is known to have a short half-life. Park and colleagues reported that the anti-inflammatory effect of the peptide is limited since the concentration required for its therapeutic effects is high [36]. Similarly, the effects observed in our in vitro experiments required a high concentration of WKYMVm peptide. Because the half-life of this peptide is known to be noticeably short, further study on the persistence of its therapeutic effects in vivo in animals is needed for its development as a therapeutic agent.

In conclusion, the WKYMVm alleviates hepatic fibrosis and promotes hepatic regeneration via vascular remodeling by FPR2. These data suggest that the WKYMVm may be a new therapeutic agent for liver fibrosis.

## 4. Materials and Methods

### 4.1. Materials

The structure of the WKYMVm peptide is shown in the Figure 1A. The synthetic peptide WKYMVm was synthesized by Anygen(Anygen, Kwangju, Korea). The purity of the synthesized WKYMVm peptide was > 98%.

### 4.2. Animals

Seven-week-old male Sprague-Dawley rats were obtained (Orient Bio Inc., Seongnam, Korea) and maintained in an air-conditioned facility. The common bile duct was ligated following anesthesia with 2,2,2-tribromoethanol (Avertin; Sigma-Aldrich, St. Louis, MO, USA). One week after the surgery, 2.5 mg/kg WKYMVm dissolved in deionized water was administered by intraperitoneal injection twice per week for two weeks. The rats were randomly assigned to one of the following groups: the normal group (Nor; *n* = 5), untreated BDL group (NTx; *n* = 20), and WKYMVm-treated BDL group (+WK; *n* = 12). All animal experimental processes were performed under a protocol that was in accordance with the guidelines of the Institutional Review Board of CHA General Hospital, Seoul, Korea. The experimental protocols were approved by the Institutional Animal Care and Use Committee of CHA University, Seongnam, Korea (IACUC-180023).

### 4.3. Serum Biochemistry Analysis

Aspartate aminotransferase (AST), alanine aminotransferase (ALT), total bilirubin and albumin (ALB) levels in the serum samples were evaluated by Southeast Medi-Chem Institute (Busan, Korea). The analysis was performed in triplicate.

### 4.4. Enzyme-Linked Immunosorbent Assay (ELISA)

The levels of VEGF and HGF were analyzed by ELISA. Concentration was quantified using human VEGF (Abcam Inc., Cambridge, MA, USA) and rat HGF (R&D Systems, Minneapolis, MN, USA) ELISA kit in strict accordance with the manufacturer’s instructions and detected using a microplate reader (BioTek, Winooski, VT, USA). All reactions were analyzed in triplicate.

### 4.5. Cell Culture

T-HSCs, immortalized rat HSCs, were maintained in α-Minimum Essential Medium (Eagle; α-MEM; GIBCO BRL, Langley, OK, USA) supplemented with 1% penicillin streptomycin (pen-strep; GIBCO BRL) and 10% fetal bovine serum (FBS; GIBCO BRL). Additionally, HUVECs were cultured in EC medium (ScienCell, Carlsbad, CA, USA) at 37 °C in a 5% CO_2_ incubator. T-HSCs and HUVECs were treated with 1 mM WKYMVm and then with TGF-β (2 ng/mL;PeproTech, Rocky Hill, NJ, USA) and 5-FU (100 μg/mL; Sigma-Aldrich) for 72 h.

### 4.6. Sirius Red Staining

Formalin-fixed rat livers were embedded in paraffin, cut into 5-μm section and stained with Sirius Red according to standard procedures [37]. The area of collagen deposition, as determined by the percentage of the area positive for Sirius Red, was quantified by ImageJ software from the histological images.

### 4.7. Immunohistochemistry

To analyze the degree of hepatocyte proliferation in tissues following treatment with WKYMVm or no treatment, we used an anti-proliferating cell nuclear antigen (PCNA; Santa Cruz Biotechnology, Dallas, TX, USA) antibody. The liver sections were incubated with 3% H_2_O_2_ in methanol to block endogenous peroxidase activity. After antigen retrieval, the slides were incubated with an anti-PCNA antibody diluted 1:200 at 4 °C overnight, and then incubated for 30 min with a biotinylated secondary anti-rabbit antibody at room temperature (RT). The sections were incubated with a horseradish peroxidase-conjugated streptavidin-biotin complex (Dako, Santa Clara, CA, USA) and 3,3-diaminobenzidine (EnVision Systems, Santa Clara, CA, USA) to visualize the chromatic signals. Images were taken using a digital slide scanner (3DHISTECH Ltd., Budapest, Hungary). Finally, the percentage of PCNA-positive hepatocytes was quantified in randomly selected sections (three fields per group at 400× magnification).

### 4.8. Western Blot

The liver tissues and cells were lysed in lysis buffer (Sigma-Aldrich) supplemented with Phosphatase Inhibitor Cocktail II (A. G Scientific Inc., San Diego, CA, USA) and Complete Mini Protease Inhibitor Cocktail (Roche, Basel, Switzerland). Total proteins were loaded onto SDS-PAGE gels and transferred to PVDF membranes (BIO-RAD, Hercules, CA, USA), and then incubated overnight at 4 °C with one of the following primary antibodies as indicated: rabbit anti-Col Ⅰ) (1:1000; Novus, Centennial, CO, USA), mouse anti-α-SMA (1:1000; Dako), mouse anti-VEGF (1:500; Novus), rabbit anti-ALB (1:1000; Novus), mouse anti-Cyclin D1 (1:1000; Abfrontier, Seoul, Korea), rabbit anti-hepatic nuclear factor α (HNF1α; 1:1000; Abcam Inc., Cambridge, MA, USA), mouse anti-IL-6 (1:1000; Abcam), rabbit anti-glycoprotein 130 (gp130; 1:250; Santa Cruz Biotechnology), rabbit anti-phosphorylated Stat3 (1:500; Cell Signaling), mouse anti-Stat3 (1:500; Cell Signaling), and rabbit anti-GAPDH (1:3000; Abfrontier). The membranes were washed and then reacted with a secondary antibody (horseradish peroxidase-conjugated anti-mouse IgG (1:5000; Cell Signaling) or anti-rabbit IgG (1:10,000; Cell Signaling) for 1 h at RT. The membranes were incubated using enhanced chemiluminescence reagents (Thermo Fisher Scientific., Waltham, MA, USA).

### 4.9. Quantitative Reverse Transcription PCR

Total RNA was isolated from rat liver tissues, HSCs, and HUVECs with TRIzol (Invitrogen, Carlsbad, CA, USA). Reverse transcription was performed with 500 ng total RNA and Superscript III reverse transcriptase (Invitrogen). Real-time PCR was performed using SYBR Master Mix (Roche) and the CFX Connect™ Real-Time System (BIO-RAD). GAPDH were used as internal controls for gene expression assays. The sequences of the primers are shown in Table 1. All reactions were performed in triplicate.

### 4.10. Immunofluorescence

To assess the localization of vWF and α-SMA in liver tissues, cryosections (7 µm thick) were fixed in 100% methanol (Merck) and incubated with blocking solution (Dako) for 1 h at RT followed by anti-vWF (1:200, Abcam) and anti-α-SMA (1:400, Dako) primary antibodies at 4 °C overnight. The sections were incubated with Alexa 488- and 568-conjugated secondary antibodies (1:400; Invitrogen) for 1 h at RT. To evaluate the localization and expression of α-SMA in rat HSCs, T-HSCs treated with WKYMVm were fixed with 100% methanol. The cells were incubated with blocking solution (Dako) at RT for 40 min and a rabbit anti-α-SMA (1:200) antibody at 4 °C overnight. Subsequently, the cells were incubated with an Alexa 568-conjugated secondary antibody (1:400, Invitrogen) at RT for 1 h and counterstained with DAPI. Images were taken with a confocal microscope (LSM 700) and analyzed with ImageJ software. All experiments were performed in triplicate.

### 4.11. Tube Formation Assay

To analyze the angiogenic ability of HUVECs, cells were stained with Alexa Fluor 488-conjugated Ac-LDL (Invitrogen). Then, the HUVECs (5 × 104 cells/well) were seeded in 24-well culture plates containing coverslips precoated with 5% Matrigel (Sigma-Aldrich) and cultured with or without 100 μg/mL of 5-FU at 37 °C in a 5% CO_2_ incubator. The cultured cells were washed with cold phosphate-buffered saline (PBS). The branch lengths of the tubes were measured and quantified using ImageJ software. All experiments were performed in triplicate.

### 4.12. Dextran Permeability Assay

HUVECs (5 × 10^4^ cells/well) were seeded in the upper chamber of a 24-well Transwell system and cultured for 24 h to allow formation of a confluent monolayer. The monolayer cells were treated with 100 μg/mL of 5-FU and after 72 h, the monolayer cells were treated with 1 mM WKYMVm peptide for 24 h. Dextran permeability was evaluated by the addition of 10 µL of 10 mg/mL fluorescein isothiocyanate (FITC)-dextran (Sigma-Aldrich) to the upper chamber for 30 min. After 30 min, 100 µL of the conditioned media in the lower chamber was transferred to a 96-well plate (BD Biosciences, Bedford, MA, USA), and the fluorescence levels of the samples were measured at an excitation wavelength of 490 nm and an emission wavelength of 525 nm using a fluorescence plate reader. The experiments were performed in triplicate.

### 4.13. Statistical Analysis

Data are presented as the mean ± standard error of the mean. Differences between different regions of SBEM were analyzed with GraphPad Prism software and the statistical methods used for comparison between pairs. In summary datasets with more than two groups were analyzed with one-way ANOVA. Datasets with two groups were analyzed with a Student’s t test. Significance was taken as * *p* < 0.05.

## 5. Conclusions

The WKYMVm peptide improves vascular remodeling and inhibits fibrosis in a rat model of hepatic failure. Furthermore, WKYMVm enhances hepatic function by upregulating the expression of hepatic function markers. These data suggest that the WKYMVm peptide acts as a modulator of liver function as well as vascular regeneration in hepatic failure.

## Figures and Tables

**Figure 1 ijms-22-02107-f001:**
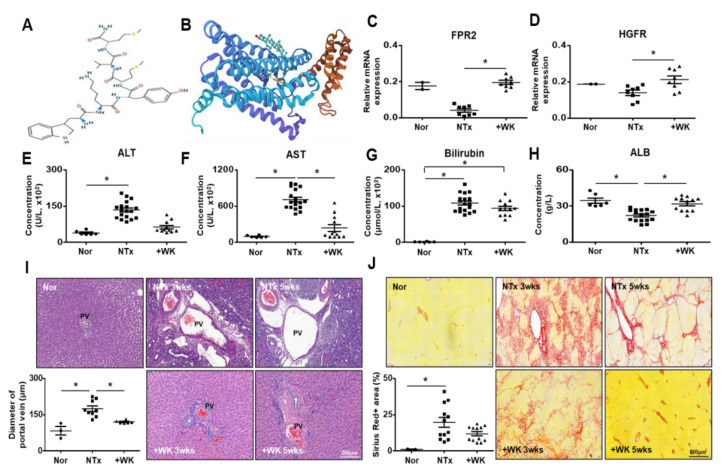
WKYMVm attenuates pathologic changes caused by bile duct ligation (BDL). (**A**) 2D structure of WKYMVm. (**B**) 3D structure of WKYMVm with FPR2. (**C**,**D**) The mRNA of FPR2 and HGFR in the livers of normal, BDL, and WKYMVm-treated BDL rats were analyzed by qRT-PCR. (**E**–**H**) The level of hepatic function markers in the BDL rat serum was analyzed. (**I**) H&E staining of liver sections. Scale bar: 200 μm. The diameter of the portal vein was quantified. (**J**) Sirius Red staining. Scale bar: 800 μm. The Sirius Red-positive area was quantified as the percentage of the total area. Mean ± SD (*n* = 3 per group), * *p* < 0.05; one-way ANOVA. Nor, normal control group; NTx, untreated BDL group; +WK, WKYMVm-treated BDL group; PV, portal vein; wks, weeks.

**Figure 2 ijms-22-02107-f002:**
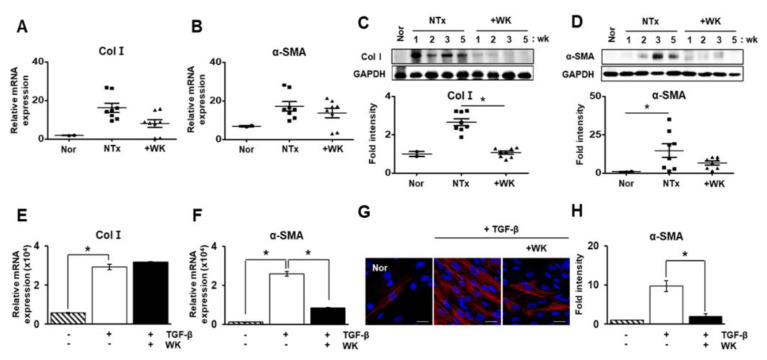
WKYMVm inhibits fibrogenic factor expression. The mRNA expression of Col I (**A**) and α-SMA (**B**) in the livers of normal, BDL, and WKYMVm-treated BDL rats was analyzed by qRT-PCR. Western blot analysis of Col I (**C**) and α-SMA (**D**) protein expression. The mRNA levels of Col I (**E**) and α-SMA (**F**) in TGF-β-treated HSCs were measured by qRT-PCR. (**G**) Immunofluorescence analysis of α-SMA in TGF-β-treated HSCs. Scale bar: 20 μm. (**H**) Staining intensity was quantified as the fold change. Mean ± SD (*n* = 3 per group), * *p* < 0.05; *t*-test or one-way ANOVA. Nor, normal control group; NTx, untreated BDL group; +WK, WKYMVm-treated BDL group; wk, week.

**Figure 3 ijms-22-02107-f003:**
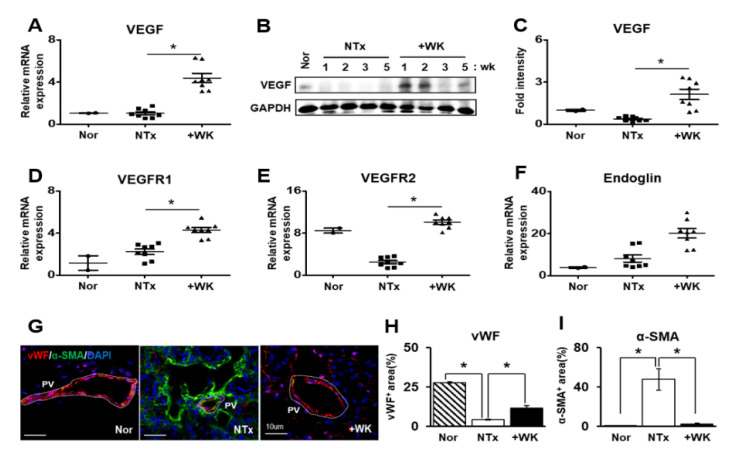
WKYMVm exerts angiogenic effects in the BDL rat liver. (**A**–**C**) The mRNA and protein levels of vascular endothelial growth factor (VEGF) in the livers of normal, BDL, and WKYMVm-treated BDL rats were analyzed by qRT-PCR and western blot analysis, respectively. The mRNA levels of VEGFR1 (**D**), VEGFR2 (**E**), and Endoglin (**F**) were measured by qRT-PCR. (**G**) Immunofluorescence staining with antibodies against vWF (red) and α-SMA (green). Quantification of vWF (**H**)- and α-SMA (**I**)-stained cells. Mean ± SD (*n* = 3 per group), * *p* < 0.05; one-way ANOVA. Nor, normal control group; NTx, untreated BDL group; +WK, WKYMVm-treated-BDL group; PV, portal vein; wk, week.

**Figure 4 ijms-22-02107-f004:**
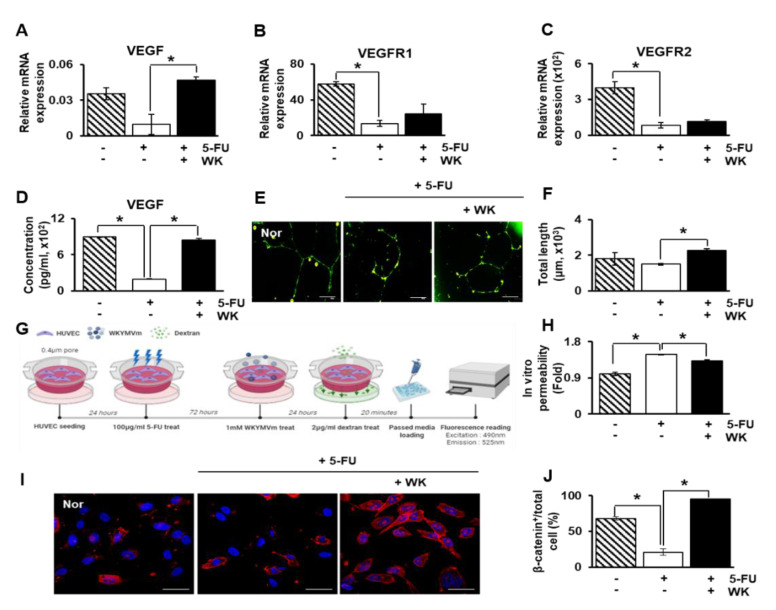
WKYMVm exerts angiogenic effects in human umbilical vein endothelial cells (HUVECs). The mRNA levels of VEGF (**A**), VEGFR1 (**B**) and VEGFR2 (**C**) in HUVECs pretreated with 5-FU and treated with WKYMVm were analyzed by qRT-PCR. (**D**) VEGF secretion by HUVECs was measured by ELISA. (**E**) Tube formation assay. Scale bar: 20 μm. (**F**) Quantification of the tube length in micrometers. (**G**) Experimental scheme of the dextran permeability assay. (**H**) Quantification of permeability was determined by the fluorescence intensity in the lower chamber. (**I**) Immunofluorescence analysis of active β-catenin in HUVECs. (**J**) Quantification of the percentage of nuclear β-catenin. Mean ± SD (*n* = 3 per group), * *p* < 0.05; *t*-test. Nor, normal control group; NTx, untreated BDL group; +WK, WKYMVm-treated BDL group.

**Figure 5 ijms-22-02107-f005:**
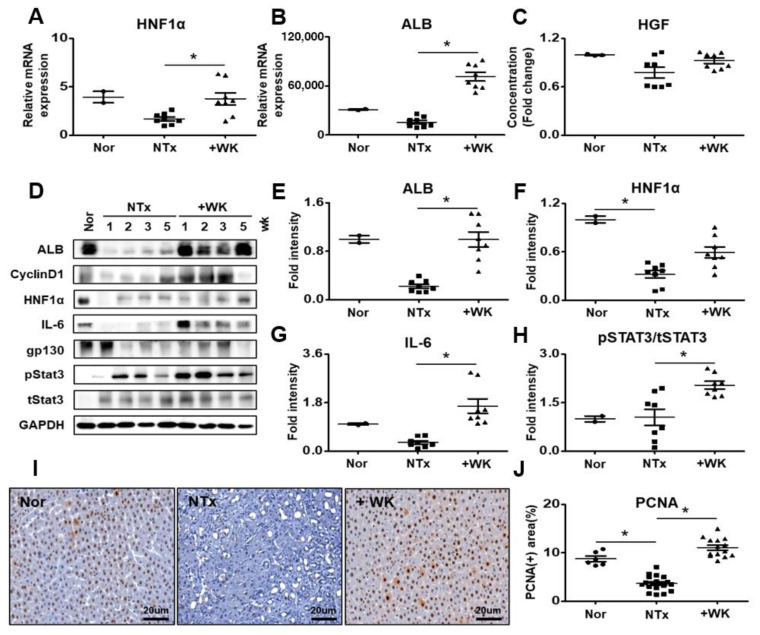
WKYMVm promotes hepatic regeneration in the BDL rat model. The mRNA levels of hepatocyte nuclear factor 1α (HNF1α) (**A**) and albumin (ALB) (**B**) in the livers of normal, BDL, and WKYMVm-treated BDL rats were analyzed by qRT-PCR. (**C**) The serum level of HGF was determined by ELISA. (**D**) The protein expression of ALB, CyclinD1, HNF1α, IL-6, gp130, pSTAT3 and tSTAT3 was determined by western blot analysis. The intensity of western bands such as ALB (**E**), HNF1α (**F**), IL-6 (**G**), and STAT3 (H) normalized by GAPDH. (**I**) Immunohistochemistry of PCNA in liver sections. (**J**) Quantification of the percentage of PCNA-positive cells. Mean ± SD (*n* = 3 per group), * *p* < 0.05; one-way ANOVA. Nor, normal control group; NTx, untreated BDL group; +WK, WKYMVm-treated BDL group; wk, week.

**Table 1 ijms-22-02107-t001:** Primer sequences using quantitative real time polymerase chain reaction.

Gene	Sequence	Accession Number
FPR2	F: 5′-ACTGTTGAAGAAGTGCTGGA-3′	XM_001073508.5
	R: 5′-AACAAGTGCTCTTCTGTGGA-3′	
HGFR	F: 5′-ACCCAGATTGTTTTCCTTGT-3′	XM_032906920.1
	R: 5′-ATTGTCAGGAGGAAGGACAT-3′	
VEGF	F: 5′-ACGGACAGACAGACAGACAC-3′	NM_001287114.1
	R: 5′-CTTCTGGGCTCTTCTCTCTC-3′	
VEGFR1	F: 5′-CCACACCTGAAATCTACCAA-3′	NM_019306.2
	R: 5′-TGGGGACTGAGTATGTGAAG-3′	
VEGFR2	F: 5′-AAGCAAATGCTCAGCAGGAT-3′	NM_013062.1
	R: 5′-TAGGCAGGGAGAGTCCAGAA-3′	
Endoglin	F: 5′-AAGGTGTGACTGTACACAAG-3′	NM_001010968.2
	R: 5′-CCAGATCTGCATATTGTGGT-3′	
Col Ⅰ	F: 5′-CATGTTCAGCTTTGTGGACC-3′	NM_053304.1
	R: 5′-GCAGCTGACTTCAGGGATGT-3′	
α-SMA	F: 5′-AACTGGTATTGTGCTGGACTCT-3′	NM_031004.2
	R: 5′-CTCAGCAGTAGTCACGAAGGA-3′	
HNF1α	F: 5′-AAGATGACACGGATGACGATGG-3′	NM_012669.1
	R: 5′-GGTTGAGACCCGTAGTGTCC-3′	
ALB	F: 5′-CTT CAA GCC TGG GCAGTAG-3′	XM_032916218.1
	R: 5′-GCACTGGCTTATCACAGCAA-3′	
GAPDH	F: 5′-TCCCTCAAGATTGTCAGCAA-3′	NM_017008.4
	R: 5′-AGATCCACAACGGATACTT-3′	

## Data Availability

Not applicable.

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
