# Peer review of "Formyl Peptide Receptor 2 Alleviates Hepatic Fibrosis in Liver Cirrhosis by Vascular Remodeling"

_ijms, 2021, doi:10.3390/ijms22042107_

Round 1

Reviewer 1 Report

The manuscript by Jun and colleagues entitled " Formyl peptide receptor 2 alleviates hepatic fibrosis in liver cirrhosis by vascular remodeling" demonstrated the effect of WKYMVm in BDL model. The manuscript should be significantly improved upon consideration of the following points:

- It needs to describe about the relationship between HGFR and WKYMVm in the introduction section before mentioning HGFR in the results.

- Fig. 1C-H and IJ, more statistical analyses are required (e.g., Nor vs NTx, Nor vs +WK, NTx vs +WK). And also for other figures if necessary.

- Does the diameter PV in Fig. 1I indicate the analysis of staining sections of samples procured at 3wks or 5wks? Result description “~depended on the degree of fibrosis” is not that much clear without showing quantification. Also, same thing can be applied for a dot plot in Fig. 1J.

- Data interpretation as well as descriptions for Fig 2E-H are not clear. Is TGF beta-induced Col I also affected by WKYMVm?

- More data descriptions for Fig. 3 are required. For example, VEGFR2 is downregulated in NTx compared to Nor. Also, is there no change of VEGF (mRNA and protein levels), VEGFR1, and endoglin levels between Nor and NTx?

- Is there any specific reason to use 5-FU for Fig. 4? Treatment with 5-FU is associated with BDL model?

- Lines 187-189: need to revise ALL sentences. Also, “increased tendency” is not a clear term. Any statistics?

- How is it feasible to mention that WKYMVm enhances angiogenic functions based on Fig 4H?

- Discussion: repeating too many data descriptions without any new insights. A paragraph (289-301) is difficult to understand. Overall the discussion needs much more stringency and a conceptual,

- Line 110: delete (Figure 2, A and B)

- Line 113: Figure 2, B and C?

- Line 125: (HSC)s is a typo. Also, check any other typos throughout the manuscript.

- Fig 1I legend, indicate abbreviation of portal vein.

- Fig. 5D: is phospho-gp130 detection available?

- Indicate PV in Fig 3G

- full name of HSCs in abstract

full name of LSEC

Author Response

Dear Editor,

We greatly appreciate your careful evaluation of our manuscript (IJMS-1100341) entitled: Activation of formyl peptide receptor 2 alleviates hepatic fibrosis in a rat model of bile duct ligation by stimulating vascular remodeling.” We also thank you for your patience because our revision took somewhat more time to address all issues raised by the reviewers.

We were really encouraged by the reviewers’ positive comments and constructive suggestions.

I am happy to report that we have successfully addressed all issues and concerns through additional data and subsequent revision of our manuscript, as detailed in the following response page. Changes are highlighted in red in the revised manuscript

Reviewer 2 Report

I’ve read with attention the paper of Kim et al. that is potentially of interest. The background and aim of the study have been clearly defined. The methodology applied is overall correct, the results are reliable and adequately discussed. I’ve only some minor comments: - From a statistical point of view I doubt that all the data compared by the authors had normal distribution so that the statistical tests should be adapted

- The authors should enrich the discussion with some more comments on their study limitation and their study perspectives.

Author Response

(The authors gave the same response as above.)

Reviewer 3 Report

This is an interesting study on the formyl peptide receptor 2 and its activation by the hexapeptide WKYMVm in the amelioration of hepatic fibrosis. The major concerns are whether the FPR2 directly influences the effects WK directly. Why wasn't a knock-out mouse model of FPR2-Ko used in your studies and a know-down of FPR2 in the in vitro studies. The other major concern is there no information on liver function in the alleviation of hepatic fibrosis by WK+. You may have pathological and molecular changes that indicate the resolution of fibrosis, but lack of liver function. It would have been appropriate to measure levels of protein, cholesterol, glucose, fatty acids, and ammonia levels in your models after administration of WK+. The background information can also be improved with respect to association with humans and the function of the FPR2 in signal transduction pathways.

Author Response

(The authors gave the same response as above.)
